



# UFlow 1.0: A Computer Model for Projections of Urban Sprawl

André Koscianski[1]

[1]UTFPR, Federal Technological University of Paraná, Av. Monteiro Lobato km 4, Ponta Grossa, Paraná, Brazil

**Correspondence:** André Koscianski (koscianski@utfpr.edu.br)

**Abstract.**

Cities concentrate most of the world's population and are the stage of difficult problems around logistics, economy, or quality of life, to enumerate just a few. As an object of research on itself, an urban agglomeration is difficult to characterize; it is both an ensemble of various disconnected heterogeneous elements, and the product of numerous actions and effects between those elements. Studies of the structure and the functioning of cities date back to one century ago, with an increased interest in the last decades on the phenomenon of expansion and all of its impacts.

Models of city growth face the complex nature of this system and are approximate. Different representations seek to balance characteristics as data availability, level of detail of internal processes, or precision. The uflow model approaches the problem with the metaphor of an abstract field, which evolves over time and signals the conversion from empty to urban cells. The procedure for calibration adjusts parameters according to the history of the region under study, and is able to capture local conditions. The implementation takes advantage of parallel hardware, and the simulation can be performed in reverse mode, a feature that can be useful to verify the adaptation of the tool to a given scenario, or to compute approximations of the past state of a region.

Tests confirmed the expected behaviour of the algorithms, and good agreement with actual data. The flexibility of the concept of intensity of urbanization is open to the integration of different data sources into the model, and the possibility of simulating their evolution over time.

## 1 Introduction

The expansion of urban areas in many locations of the world is a disorganized process. This brings a series of negative consequences, both for the environment and for communities trapped into anarchic systems: the list of problems is extensive, with examples as floods and erosion, soil and air pollution, overloaded transport systems and risky, irregular constructions. Trendlines on demographic expansion and rural exodus do not point to equilibrium, at least for some decades to come; and although global awareness has improved around issues as pollution and sustainability, we still witness urban agglomerations growing larger and where, unfortunately, social, economic and ecological problems will be present with higher frequency and higher intensity. Author J. Jacobs remarkably foresaw these difficulties half a century ago: "Meanwhile, all the art and science of city planning are helpless to stern decay - and the spiritlessness that precedes decay - in ever more massive swatches or cities." (Jacob, 1992).



Projections of urban expansion are an integral part of the decision-making of legislators, politics and investors. Cities evolve according to complex sets of forces and constrains, which are difficult to observe and are unlikely to be fully managed. It is a rich process, which gave rise to several models of urban spraw.

Research involving Land Use and Cover Change - LUCC - faces a classical dilemma: refinements of representation that incorporate more details engender uncertainties and are harder to understand and control; inversely, models restricted to fewer parameters are more amenable to analysis, but may lack flexibility to accomodate complex dynamics (Rocha, 2012). The field of urban sprawl simulation encompasses different approaches and computational tools, depending on factors as data availability and model perspective (Wray et al., 2013; Musa et al., 2017).

System dynamics models, for instance, structure a simulation as three subsystems: business, housing and population (Sanders and Sanders, 2004). These models are not spatially-oriented; they are based on concepts as stocks and flows, and are able to represent nonlinear effects (Theobald and Gross, 1994; Sterman, 2002).

Spatial Economics / Econometric Models put emphasis in financial aspects, as relations between demography and housing markets (Haase and Schwarz, 2009; Wray et al., 2013). For instance, Engle (1974) makes a general analysis of models of

urban growth encompassing factors as economic activities, supply and demand of products, output of manufacturing sectors and labor force. As another example, Mankiw and Weil (1989) analyse the expansion of the housing market in the USA in relation to the baby boom during the 1970 decade.

Other techniques with analytic background and using GIS data include Logistic Regression (Hu and Lo, 2007; Alsharif and Pradhan, 2014), Neural Networks (Liu and Seto, 2008; Tayyebi et al., 2014; Pijanowski et al., 2014) and Markov chains

(Aaviksoo, 1995; Guan et al., 2011; Moghadam and Helbich, 2013). These models have a tendency of generalization and require adaptations to represent fine spatial variability (Lu et al., 2003; Pijanowski et al., 2014; Moghadam and Helbich, 2015).

Agent based models, ABM, constitute a recent category in the history of computer simulation. An agent can represent an autonomous entity, capable of interacting with other agents and the surrounding environment (Crooks and Heppenstall, 2012). ABM support a bottom-up approach, allowing to represent actions and states with fine granularity. It can be used to represent

internal city dynamics (Barros, 2012), or go as far as simulating artificial societies and how they build and modify a city (Li and Liu, 2007; Schmitt et al., 2014). The high level of detail of this technique has a drawback, which is finding the correct parameters and calibrating the model (Li and Yeh, 2001).

One of the most used techniques in the context of LUCC is Cellular Automata (CA). It is a powerful modelling concept, exhibiting complex behaviour from simple structures (Wolfram, 1984), finding applications that range from fluid dynamics

to sociological studies. CA accomodates deterministic rules and stochastic reasoning (Wu, 2002), its lattice structure is GIS-friendly and simplifies the implementation of different perspectives, as the computation in the same model of local and global effects.

This article presents a model of urban sprawl using a modifications of the classic CA: instead of discrete transition rules, the evolution of the city footprint is dictated by what might be called a potential for change, governed by an analytic model. Local

conditions are assimilated into a grid of parameters that is automatically calibrated to fit a prescribed trajectory, and a second processing step employs a probability map to generate disconnected urban fragments. The model is very transparent, in the





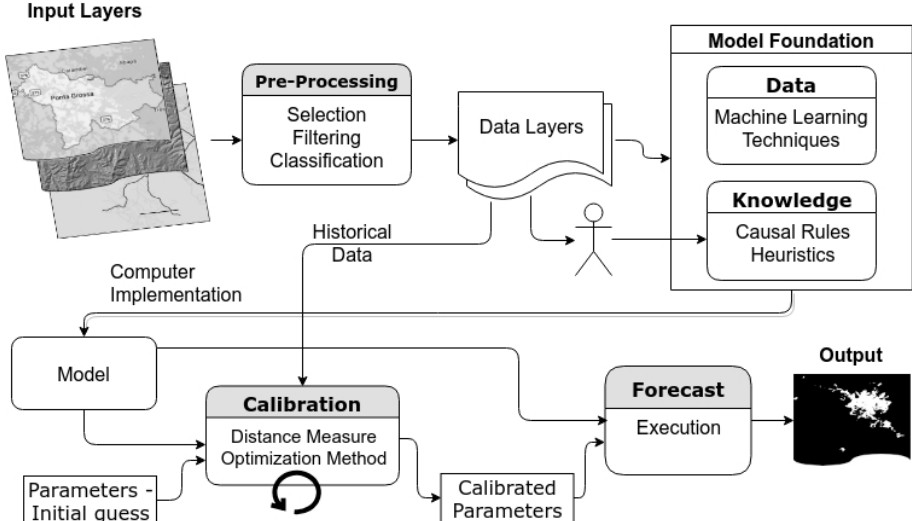

**Figure 1.** Overview of Urban simulation.

sense that outputs are linked to the inputs in a straightforward manner, and it can be used as a basis to implement other criteria governing grow speed.

## 2    Overview of CA-based LUCC simulation

LUCC simulators perform computations derived from geophysical and historical data, in order to extrapolate the future behaviour of a region. The most basic and common output is a classification, usually in Boolean form: an area is either urban or not, although there are models which break this division further into categories as residence, rural and industrial areas, or parks (Barredo et al., 2003).

To say that LUCC forecast is a sophisticated form of data regression is a fair description (Brunsdon et al. (1996) being

an example), but it misses a central feature of the problem. What's specially remarkable in the study of city dynamics is its ambivalence, as deterministic rules operate together with random events to produce results. For instance, smooth topography, transportation facilities and distance to commercial zones are examples of quantifiable information, and catalysts for terrain occupation. On the other hand, unexpected events as the shutdown of a factory (e.g.: Chapain and Murie (2008)), and investors decisions may change the future of a city (Batty, 2007; Rocha, 2012).

The mix of deterministic causal rules and random factors may point two different directions to build a representation, depicted in Figure 1.

First, the whole dynamics comprising deterministic and aleatory effects can be absorbed into black-box models, commonly referred as machine learning tools. Examples in this line include Logistic Regression (Hu and Lo, 2007), Neural Networks (Li and Yeh, 2002; Almeida et al., 2008), Stochastic Automata (Wu, 2002), Particle Swarm (Feng et al., 2011), Random Forests





(Kamusoko and Gamba, 2015), Deep Learning (He et al., 2018) or Markov models (Aaviksoo, 1995; López et al., 2001; Guan et al., 2011; Arsanjani et al., 2013; Moghadam and Helbich, 2013). The accuracy of a simulation built this way will be mainly affected by the volume of available data, the correct combination of variables in calculation procedures, and the presence of patterns that can be detected and remain valid throughout the evolution of the city.

Second, emphasis is placed on known facts about the system. This knowledge is translated into rules to describe transitions,
relations, effects, which have higher interference over the processes occurring in the system, complemented with data that backs the construction of a simulation. The rules can depict hierarchical levels and be parameterized to fit the dynamics of a particular scenario. Parameters that control the simulator are adjusted to fit the scenario under study. Because of the intrinsic stochastic nature of the problem, deterministic models are extended with sources of perturbation, to give them some organic resemblance and break rigid patterns that may fall short of representing actual behaviours. Models of this genre include Agents
(Arsanjani et al., 2013; Ligtenberg et al., 2001; Parker et al., 2003), Rule-based Systems (Berberoglu et al., 2016) and Cellular Automata (Rocha, 2012; Santé et al., 2010; Moghadam and Helbich, 2013; Berberoglu et al., 2016; Clarke, 2008).

Simulation generally takes into account the whole dataset; regional trends are averaged out, and this choice owns to the need to keep the complexity of data and algorithms under a manageable level. Alternatives to circumvent this problem consists of partitioning the map (Ke et al., 2016; Shu et al., 2017; McGarigal et al., 2008), or to structure the model in different scales
(White and Engelen, 2000; Stevens and Dragićević, 2007).

## 2.1  Cellular automata

Cellular automata executes a set of fixed rules over tabular data. The ability to exhibit complex execution patterns stems from the large set of states that a grid can represent. If the state $z$ of a cell depends on all of its neighbours (a common configuration), the transition $z^{new} = f(s^{old}_{1..8})$ can be chosen among 256 possibilities. A tiny grid with 10 x 10 binary cells (urban/non urban)
has room for $2^{100}$ situations. A thorough review of CA properties can be found in Wolfram (1983).

Simulation is carried out by firing CA rules in sequence; as an example, an initial step might freeze the state of excluded areas, then urbanisation might occur driven by proximity to town centre, industries and roads. In Besussi et al. (1998) this scheme is implemented as a cascade of CA models. From a programming perspective, rules are not fundamentally different from functions; in Barredo et al. (2003), for instance, transitions between 22 possible states are determined by a function that
computes a stochastic potential.

Rules that dictate state transitions are grounded on causality relations, elements of heuristics and probability. A typical example is Besussi et al. (1998): "A residential cell marked with value y ... [close to] central area will assume the value (y+1) in a percentage x of cases.". Transition to a state $z$ is more likely to occur in the vicinity of other cells in the same state: there is little chance that industries be set in the middle of residential areas. Accessibility by different transport modes is a catalyst for
growth and for change: the odds that an abandoned lot get used are higher if it is located near large roads. Likewise, distance to city centre is a good estimator for the utility or the level of interest of a lot of land. Topography may operate in the inverse sense: proximity to rivers and lakes may be associated with probability of floods, and steep slopes generally inhibit construction and land usage, although irregular occupations may disregard these aspects (Reuß, 2017).





This ensemble of factors is sometimes summarized as a set of attractive and repulsive forces (White and Engelen, 2000;
Stanilov and Batty, 2011; Furtado et al., 2012), and an illustrative example of this metaphor is the use of gravity equations
(Chen, 2009, 2015). Figure 2 is an illustration of forces acting in a city (Ponta Grossa, Brazil). The two locations are approximately 500 meters apart; the steep, unpaved street (Bartolomeu Bueno) is close to a flat boulevard (Carlos Cavalcanti av.) and
a hypermarket. Land prices climb along of the slope in the direction of the avenue.

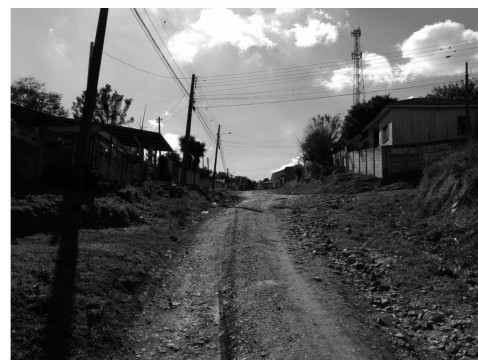
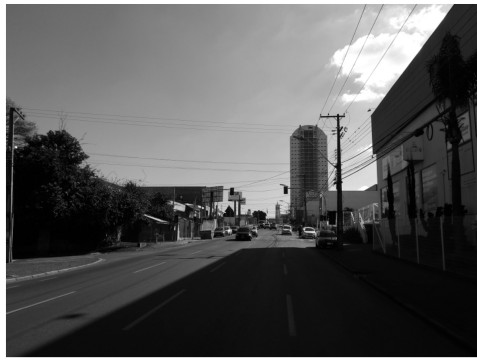

Bartolomeu Bueno st.                        Carlos Cavalcanti av.

**Figure 2.** Illustration of slope influence: two locations 500 meters apart in Ponta Grossa/Brazil.

## 2.2 Metrics and Calibration

A run of a simulator $s(\cdot)$ takes as input a two-dimensional map or matrix $m_n$, a set of parameters $\boldsymbol{\theta}$, and produces a new city
map $m_{n+1} = s(\boldsymbol{\theta}, m_n)$. Some tools require a set of maps, depicting several years. Time is generally measured in years, and
before computing any predictions, a simulator must be tuned to reproduce historical data; given a series of maps $m_{0,1,...}$, it is
expected that $m_{n+1} \approx s(\boldsymbol{\theta}, m_n)$. This entails a calibration procedure, which can be cast as an optimization problem defined
with a distance metric:

$$\boldsymbol{\theta}^* = \min_{\boldsymbol{\theta}} J = \sum_{n=1}^{N-1} \|m_{n+1} - s(\boldsymbol{\theta}, m_n)\|^2 \tag{1}$$

This formulation implies that the parameters $\boldsymbol{\theta}$ remain fixed throughout the historical series. The alternative would be to
have $\boldsymbol{\theta} = \boldsymbol{\theta}(t)$, leading to a control problem; no urban simulator, to our knowledge, operates this way. The model SLEUTH
modifies parameters during execution, but it does so according to internal heuristics and not following a prescribed trajectory
(Clarke and Gaydos, 1998).

The choice of the norm in Equation (1) is of special importance, as it may reflect different properties of the landscape (McGarigal and Marks, 1995; Amiri et al., 2017). One of the most obvious choices is the difference between simulated and actual
growth, computed as the number of urban pixels. This metric has a low computational cost, but is blind to city morphology:





expansion predicted in the right or wrong places may yield the same values. This calculation is, nonetheless, useful as a raw estimate to filter configurations during calibration. Projections of population growth (Stevens and Dragićević, 2007) enter the

same category. A comprehensive set of metrics is found in McGarigal and Marks (1995), including, for example, patch size and count, average and total perimeter of patches, and shape analysis. Research in protein structure classification also deal with shape matching with similar metrics (Baldi et al., 2000). The use of a set of measurements might provide a multifaceted view of city growth (Schneider and Woodcock, 2008), but integrating several functions in (1) would require the definition of an aggregation index (e.g. Dietzel and Clarke (2007)) and induces the problem of fixing the relative weight of each factor.

One of the basic metrics in this context is the Lee-Sallee symmetric difference (Lee and Sallee, 1970), which addresses the limitation of the simple difference. It is computed as $1 - |A \cap B|/|A \cup B|$, where $A$ and $B$ are images being compared, the union and intersection operators are computed per pixel, and the norm operator stands for pixel count.

The use of a binary classification produce four possible results: a pixel can be correctly or wrongly predicted to be urban or not. This leads to the Matthews correlation coefficient (Matthews, 1975); it combines the counts of the four types of pixels (TP

true positive, TN true negative, FP false positive, FN false negative) into a uni-dimensional measure given by:

$$\|m\| = \frac{TP \times TN - FP \times FN}{\sqrt{(TP + FN)(TP + FP)(TN + FP)(TN + FN)}} \tag{2}$$

Matthews coefficient gives more informative than Lee-Salee (Chicco, 2017), and it has been extended to compare more than two categories (Gorodkin, 2004). Another metric is the Kappa index, but its interpretation is non trivial (Pontius Jr and Millones, 2011; Olofsson et al., 2014). In the case of urban sprawl a perfect match is of lesser importance if, in a series of

snapshots, a model succeeds to reproduce general trends for a city.

Beyond quantitive metrics, calibration may incorporate analysis of experts and stakeholders of patterns, behaviours and information that may fall out of the reach of numerical scales; some examples are Sudhira et al. (2004) and Houet et al. (2016).

Simulators generally have internal parameters that control operation. Two examples are Besussi et al. (1998), where a cell can be converted to comercial usage, if surrounded by at least four residential neighbors; and Clarke and Gaydos (1998), with three

urban cells as a threshold to activate a type of growth called as organic. The choice of such values is part of the construction of a model, as registered in this rare account: "Settings for these four constants were arrived at by examining growth rates of cities over time, but are otherwise the result of trial and error" (Clarke, 2008). Tools as SLEUTH make a handful of parameters accessible to end-users (Clarke and Gaydos, 1998), but there are simulators that increase this number to dozens of adjustable coefficients (Fang et al., 2005; Arsanjani et al., 2013). Some authors go one step further, and use machine learning techniques

to find the very rules that are going to be executed (Li and Gar-On Yeh, 2004; Li and Liu, 2006; Bone and Dragićević, 2009; Feng et al., 2011).

Calibration of parameters tends to be a lengthy procedure, involving the repeated execution of the software over a large search space (e.g. Newland et al. (2018); Paegelow et al. (2014)). The presence of non-determinism make things worse, pointing the use of Monte-Carlo techniques (Clarke and Gaydos, 1998). A striking example of processing cost is presented by Schmitt et al.

(2014), with a simulation that is executed hundreds of millions of times.





Given the non-convexity and discontinuities of the optimization problem (1), techniques as gradient descent are not the most indicated. Methods commonly employed include Genetic Algorithms, Simulated Annealing, or even brute force (Newland et al., 2018; Paegelow et al., 2014; Dietzel and Clarke, 2007; Al-Ahmadi et al., 2009). Meta-heuristics have a potential interest here, but the recent flood of proposals in this area may shadow algorithms that are really effective; a thoughtful discussion in
this sense can be found in Sörensen (2015).

## 3    The UFlow model

The advance of city boundaries over empty regions is modulated by factors that have been described as attractive and repulsive forces (White and Engelen, 2000; Stanilov and Batty, 2011; Furtado et al., 2012). In the present study city sprawl is viewed under the metaphor of an urbanization flow $\nabla u(x,t)$. The proposed method does not try to explain micro-processes that take
part on expansion, and seeks instead to reproduce observed trends in the expansion of the city footprint. A similar reasoning can be found in Tayyebi et al. (2014).

Empty areas consume a flow of investments in the form of installation of infrastructures, construction of houses and buildings; resources are also spent with transportation costs, material and workforce. Once a new area is settled, it acquires a new value that, on its turn, is also a potential motor of urbanization. At some point the urbanization flow may reach equilibrium,
during which finantial resources are steadly applied in the maintenance of infra-structures. If population remains constant and there are no new investments on commerce and industry, city expansion may stall.

The flow metaphor is applicable in the inverse phenomenon of shrinkage. The decline of a city is marked by the same complexity as expansion and both situations are remarkably tied to economy; deindustrialization is one of the reasons for a city to contract (Beauregard, 2009; Haase et al., 2014; Pike et al., 2016), and this process usually begins at urban outskirts towards
the centre.

During expansion, agricultural lands and empty areas tend to be converted by the pressure exerted by the city; examples of this effect are the rise in prices in the interface between urban and farmlands (Chicoine, 1981; Plantinga et al., 2002; Livanis et al., 2006); and the impacts of the construction of a skyscraper, which irradiates an increase in land price, boosts installation of amenities and commerce, and augments the demand for infrastructures (Ahlfeldt and McMillen, 2018). All these characteristics
can be subsummed in the concept of an urbanization index $u(x,y)$, which can be viewed as a generalization of the simple binary classification of a cell as urban or empty.

The value of $u(x,y,t)$ may change over time, as a result of installation of infrastructures, renewal of a neighborhood, new constructions. Periods of decline may also happen, caused for example by lack of maintenance, physical deterioration, or severe weather events. These processes accumulate a difference in urbanization between two moments in time, in a given area. In the
one dimensional case and for small values $\Delta x, \Delta t$, the amount corresponds to

$$[u(x,t+\Delta t)-u(x,t)]\Delta x \tag{3}$$





Improvements in an area tend to propagate and ultimately lead to urban sprawl; this can be assimilated into a directional flux that drives the evolution of a region. Similar to an energy gradient (Torrens and Alberti, 2000), it can be written as

$$\Phi(x,t) = \kappa \frac{u(x+\Delta x, t) - u(x,t)}{\Delta x} = \kappa \frac{\partial u}{\partial x}(x,t)$$

, where $\kappa$ captures conditions, like slope, that interfere with the intensity of the process.

It can be shown that this expression gives rise to the following equation:

$$\frac{\partial u}{\partial t} = \kappa(x,y)\left(\frac{\partial^2 u}{\partial x^2} + \frac{\partial^2 u}{\partial y^2}\right) \tag{4}$$

This expression is known as the Heat Equation (a derivation is provided in Appendix C). It relates the change of temperature with the transfer of heat energy from hotter to colder regions. This differential equation has already been considered to represent

distribution of population (Tobler, 1970; Chen, 2008; Rocha, 2012).

The model takes two inputs: (i) a map $m_1$ of pixels corresponding to the current city map; (ii) a table $\kappa$ describing what might be called urban diffusivity. Before solving the model numerically, some adjustments to the heat analogy are necessary; this is discussed in the following section.

### 3.1 Variations on a theme

A limitation of the model (4) is the requirement that regions be adjacent to conduct flow; new zones may emerge as satellites around the main footprint. Installation of industrial plants is a typical case, and complex landforms also produce fragmented morphologies. In the present case a probability map is computed to control this mechanism.

Input maps $m_1$ and $m_2$ use two extreme values to classify land. In the present study, 0 corresponds to empty, and the value 5 was assigned to urban cells. The higher the initial value, the fastest the progression of urban borders. The change from an

empty pixel to urbanized was empirically fixed at the threshold $u > 0.5$. Different combinations of these values and $\kappa$ controls the reach of the urbanized region under the heating curve. This point is illustrated in Figure 3.

The use of binary maps flattens the city and disregards initial urbanization gradients, leaving the computation of the coefficients $\kappa$ alone to capture city dynamics. An alternative to explore is the use of information as population density, land value, distance to commercial centres and other factors (Tobler, 1969; Meentemeyer, 1989; Tayyebi et al., 2014). Since this

corresponds to a change in the initial conditions of the problem, it may affect how the solution pushes the borders of the city.

Calibration requires maps in two different points of time $m_{1,2}$, separated by an arbitrary period $t_A$. The algorithm searches the optimum $\kappa^*$, guided by Matthews' coefficient; and it also finds the simulated time $t_S$ that takes the system from $m_1$ to $m_2$.

Finding parameters that match a given trajectory corresponds to an inverse problem; the heat equation is particularly difficult in this respect, and the conditions imposed here, as discontinuities and the large number of unknowns $\kappa(x,y)$ make it harder

(Beck and Arnold, 1977; Tervola, 1989; Alifanov et al., 1995). Since the objective was only determining if the values of cells fall within a given range, an interactive algorithm was devised; in short, an error map is used to modify $\kappa^*$ in multiple iterations of the equation (4).





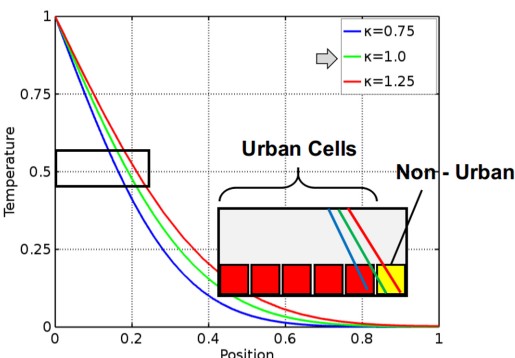

**Figure 3.** Grid resolution and reach of the urban frontier.

## 3.2 Main algorithm

Numerical solutions to the Heat Equation are available in many textbooks and won't be discussed here. The implementation
used a finite difference, forward scheme. The mesh spacing has no real physical meaning (the calculation does not represent
the heating of a plate a few kilometers large), and the values of $\Delta x$ and $\Delta y$ were set to 1, with a time step $\Delta t = 0.125$. The
relation between simulated and actual time is established at the end of the calibration procedure, according to the number of
iterations of the algorithm.

The calibration process is described by Algorithm 1. It begins by taking the initial city map $m_1$ as a constant heat source
and an initial coefficient map $\kappa(x, y) = 1$; exclusion areas as parks can be defined by setting $\kappa(x, y) = 0$.

---

**Algorithm 1** Calibration algorithm

---

j = 1; $m_S = m_1$; (initial simulation output)

**repeat**

    **while** (count of urbanized pixels in $m_S$ is lesser than in $m_2$) **do**

        Integrate (4) and advance simulated time $t_S$

    **end while**

    Compute $r = m_S - m_2$ to find pixels that were wrongly urbanized

    Update the diffusion layer, $\kappa_{t+1} = [\kappa_t - \gamma * \ell_{j*\tau}(r)]^+$ (description in the text)

    Compute Matthews' coefficient $\|m_S - m_2\|$

    j = j + 1

**until** j = $n_C$ or Matthew's coefficient ceases to improve.

Keep best $\kappa$ and return the corresponding $t_S$

---





The inner loop is iterated until the number of pixels in the urban footprint is equal to, or larger than the reference image $m_2$. The difference $r = m_s - m_2$ is calculated to obtain cells wrongly classified as urban. The minus operator in this formula takes two maps, applies the threshold $0.5$ to classify pixels as urban or not, and then for each pixel computes $(m_s(x, y) - m_2(x, y))^+$. The result is an error image containing spots that are false positives.

Next, spots in the error image $r$ are enlarged to create regions where the diffusion coefficient will be diminished, in order to form barriers that slow down the urban flow. The resulting image is subtracted from $\kappa$ and negative values are set to zero. At each iteration the error image is recalculated and the enlargement factor is increased. The effect is similar to plating a surface with a series of concentric circles of increasing radius; the inner region will have the lowest values for the diffusion coefficient.

Expanding the spots is not the same as rescaling the image; the closest morphological operation is dilation, but it deforms

borders according to a parameter known as structuring element (Haralick et al., 1987). In the present case, better results are obtained using equation (4) itself with $r$ as input, and this process will be referred here as $\ell_T(r)$ for a given $T$ interval. This appears in the Algorithm 1 in the expression $[\kappa_t - \gamma * \ell_{j*\tau}(r)]^+$, which is controlled by two parameters: $\gamma$ defines how fast insulation areas build up in $\kappa(x, y)$; and $\tau$ controls the rate of expansion of spots at each iteration. Values were chosen empirically, with $\tau = 5$ and $\gamma \approx 0.075$; the latter parameter was exposed in configuration files. These coefficients do not alter

the mechanics of the model, but influence the speed and quality of calibration.

The calibration stops if the maximum number of iterations $n_C$ is attained, of if Matthew's coefficient does not improve between cycles. The algorithm outputs $t_S$, the simulated time separating images $m_1$ and $m_2$; $\kappa^*$, a sub-optimal diffusion map; and the best value of Matthews' coefficient.

Forecasts are calculated in two steps: expansion of the second map $m_2$ followed by the generation of new urban islands.

Users define the number of years of projection, and a linear regression extrapolates the number of new pixels along of $m_1$ and $m_2$. The result is divided between expansion of existing areas and creation of new clusters, following the same proportions found in the input images.

The process of diffusion produces isotherms and creates regular, rounded shapes; to counteract this effect, the map $\kappa^*$ is modified by adding a random value $\chi$ to every pixel. Exclusion areas can be defined by assigning pixels with the value $0$.

Expansion is computed taking as inputs $\kappa^* + \chi$ and $m_2$ as a heat source with temperature set to $5.0$, and integrating Equation (4) until the growth attain or surpasses the expected count of pixels. Next, new urban fragments are created using a random spraying procedure described in the following sections; it involves analysis of the images to count urban clusters and their size, and the calculation of a probability map $\pi(x, y)$.

### 3.3    Count of satellite regions

The size and number of urban islands is important to characterize the morphology of a city and to parameterize the spraying process. The usual solution to find clusters involves edge detection techniques, but since the images are monochromatic, a simpler procedure was devised. A variable $n_R$ is initialized to 1; the image is scanned pixel by pixel and whenever a cell with value 1 is found, a flood-fill algorithm is applied to paint the region with colour $n_R + 1$. The value of $n_R$ is incremented and





the scanning proceeds until the end of the image. As the filling routine runs it also counts the number of painted cells to obtain
measures of areas.

This procedure is executed with images $m_1$ and $m_2$. The number of fragments in the images, $n_1$ and $n_2$, is used in a linear interpolation in the forecast. The area of the new clusters is chosen as an average of the $(n_2 - n_1)$ smaller clusters in $m_2$.

### 3.4 Probability map and generation of clusters

The emergence of new fragments is more intense near roads and in the proximity of the city boundaries. The effect can be related to Tobler's Law (Tobler, 1970; Miller, 2004), and depends on the distance between elements. Despite the simplicity of the concept, an efficient implementation is non-trivial (Grevera, 2007; Fabbri et al., 2008; Friedman et al., 2018; Felzenszwalb and Huttenlocher, 2012). Giving an implementations of the distance transform $\delta(\cdot)$, a probability map can be computed as

$$\pi = \delta(\rho \cup m_2) - e - (\rho \cup m_2) \tag{5}$$

, where $\rho$ is a map of roads and $e$ is an exclusion layer, both represented as binary matrices. The rightmost term removes pixels that are already urbanized. The values of $\pi$ are normalized in the interval $[0; 1]$.

A map calculated this way exhibits a linear profile, and may differ from the way the spatial dimension affects geographical variables (Torrens and Alberti, 2000). Alternatives are exponential decay (Lenormand et al., 2016) or a power law (Chen, 2015). An exponential decay can be applied as $e^{-g(1-\delta(x,y))^2}$, increasing the density of pixels generated near city boundaries.

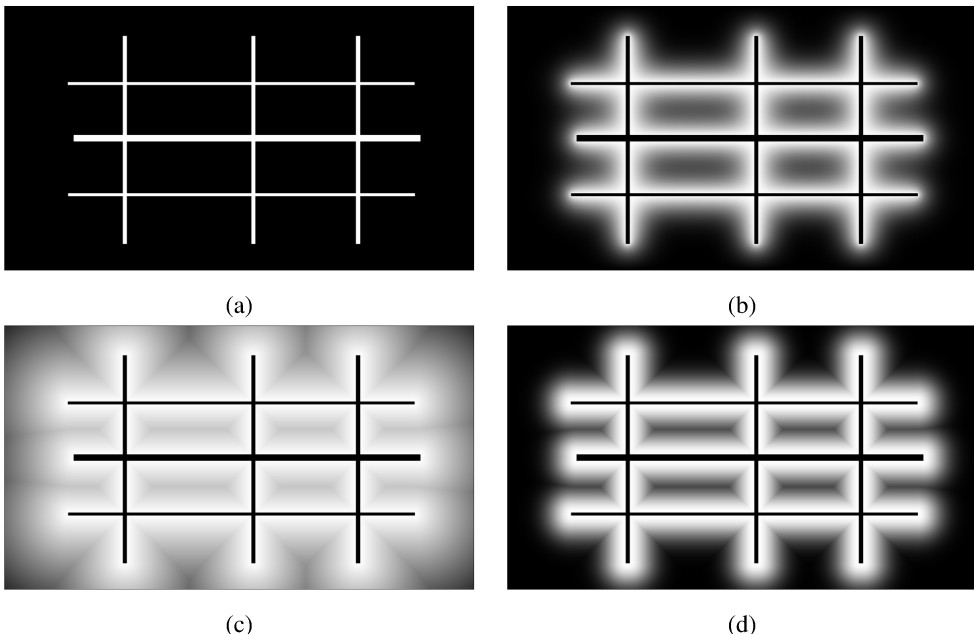

**Figure 4.** Calculation of probability maps for the spraying process: (a) road map $\rho$; (b) output of $\ell_{10000}(\rho)$; (c) output of $\delta(\rho)$; (d) exponential decay applied to $\delta(\rho)$.





The distance transform has a limitation, as it does not represent the cumulative influence of several sources. This effect is
present in heat model $\ell_T(\cdot)$, which can replace $\delta(\cdot)$ in the calculation of Equation (5). Maps generated using the two techniques
are shown in Figure 4. The results obtained with $\ell_T(\cdot)$ are more plausible for geometries as crossroads. In view of this, the
diffusion equation was chosen for the calculation of $\pi(x,y)$.

The map $\pi$ can be further adjusted with the negative influence of a slope map. Other aspects as historical and sociological
characteristics can be incorporated by manual edition (e.g. Houet et al. (2016)).

The spraying procedure follows a classic generate/reject logic, as shown in Algorithm 2.

---

**Algorithm 2** Spraying process (generation of disconnected fragments)

---

Take as input the simulated projection map $m_S$ and the probability map $\pi$

**for** $j = 1..n_R$ **do**

    Draw random value $v \in [0.1; 0.9]$ from an uniform distribution

    **for** $k = 1..100$ **do**

        Choose at random a pixel location $(x,y)$

        **if** $v > \pi(x,y)$ **then**

            Fill a rectangular region in $m_1$ with 1

            Assign zero to the corresponding pixels in $\pi$

            Exit loop

        **end if**

    **end for**

**end for**

---

The interval $[0.1; 0.9]$ in the algorithm was chosen to escape from two extreme cases. Low values for $v$ would select locations
of low relevance, while values close to 1 would place new clusters touching areas already occupied.

## 4 Verification and Validation

Conceptual model validation is the first step in checking a simulator; it assesses that "...theories and assumptions underlying
the conceptual model are correct and that the model representation of the problem entity is reasonable for the intended purpose
of the model" (Sargent, 2013). The UFlow model incorporates several ideas developed in other studies; important points
mentioned in the preceeding sections include:

– the city map is sampled in a bidimensional regular mesh, with arbitrary resolution;

– urban boundaries may expand with different speeds and irregular shapes;

– disconnected urban fragments may emerge at random moments and places;

– new urban areas are potentialized by accessibility (roads) and proximity to the main city footprint;





- exclusion areas can be arbitrarily defined;

- a probability map can represent trends of land conversion.

An important hypothesis of the model is the capability of adjusting the coefficients $\kappa$ to capture local tendencies of sprawl.

Initial operational validation (Sargent, 2013) was conducted using artificial images, allowing controlled observations similar to the ones found in N. Gazulis (2006). This test confirmed expectations as is presented in Appendix B.

### 4.1    Tests with actual maps

Two cities were tested: Ponta Grossa, Brazil, for which comparison data was available; and the capital of México, México City.

Putting aside differences of size, history and economy, the cities share two characteristics that make them interesting test-

cases. First, a significant part of urban sprawl occurred outside the control of legislation or public planning (Duhau, 1988; Silva, 2013). Second, both cities have a complicated topography, with México having its own extinct volcano, Xaltepec. These factors create more opportunities for heterogeneous local conditions and give rise to irregular morphology.

### 4.1.1    Ponta Grossa

Ponta Grossa occupies approximately $170 km^2$ and has a population density around $160/km^2$. If compared to other simi-

lar brazilian cities of the region (Paraná state), it has a very scattered layout. The city was shaped by irregular occupation, complicated topography and a lack of infra-structures as viaducts.

Reference maps for Ponta Grossa represented the years 1984, 1993 and 2002, with a resolution of $1242 \times 1339$ pixels. Images are shown in Figure 5.

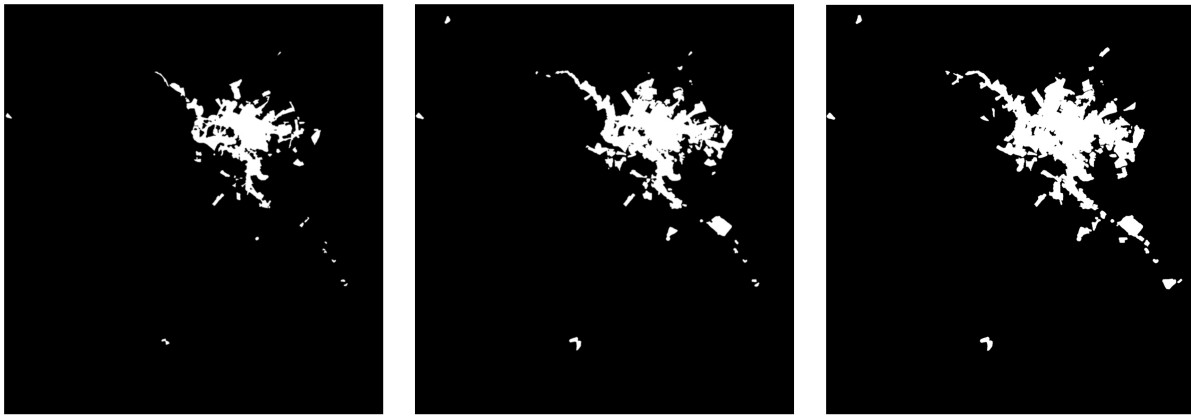

**Figure 5.** Ponta Grossa footprints; from left to right: 1984; 1993 and 2002. Images from Roth (2019).

Population augmented from approximately 173000 in 1984 to 267000 in 2002 (Prieto, 2010), a change of $54\%$. The number

of pixels in the corresponding images are 50969 and 102292, an increase of $100\%$, although a certain amount of noise is always


present in satellite images (Manandhar et al., 2009; Roth, 2019). During the simulated period, urban zoning was modified and farms were absorbed into city boundaries, but many areas remained unoccupied for reasons as litigation and lack of investments.

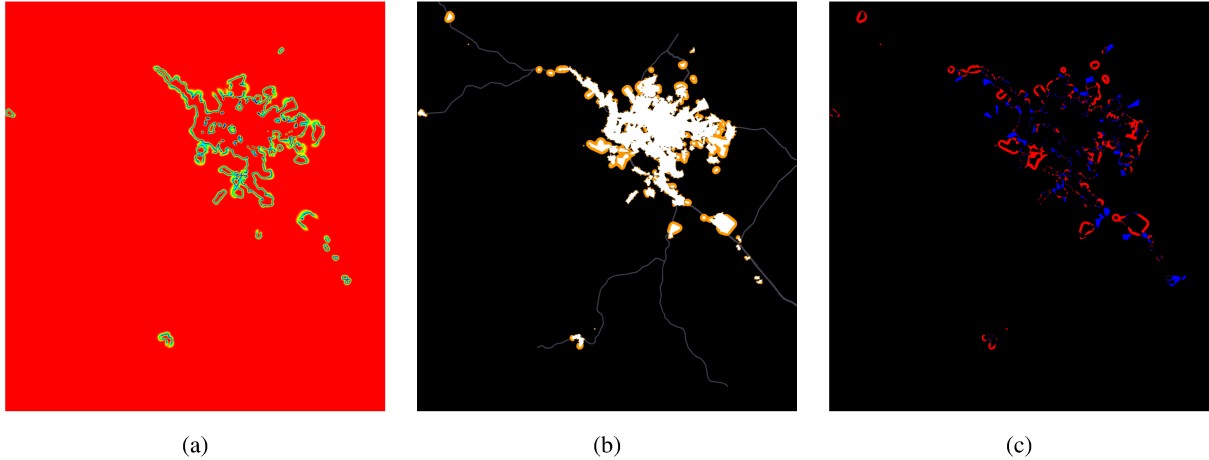

(a)                                                (b)                                                (c)

**Figure 6.** Results of calibration and forecast computed by the simulator: (a) $\kappa^*$; (b) Forecast 2002; (c) Error image.

The first part of Figure 6 shows the diffusion coefficients $\kappa^*$. Most of the area is covered in red, representing the maximum value (1.0); low values form a thin line following the contour of the city, but at some points the barrier is weaker and allows the passage of the flow. Internal regions that correspond to parks were correctly identified and insulated. The linear regression computed by the simulator projected 110151 cells in 2002, an excess of $7\%$ compared to the actual image. During the period 1984-1993 the simulation found 3 new clusters, but the actual number is slightly greater, since the heristic used to count clusters misses fragments that fuse together.

Red and blue pixels in the error image correspond respectively to false positive and false negative. The rapid alternance between the two types of errors occurs along all of the highly indented border. Images of the city in the following years confirm this growth pattern.

Figure 7 shows the footprint 15 years later in relation to the forecast: it can be seen that the city borders advance over the false positive/negative pixels of the error image, and that the expansion does not concentrate on specific regions. The second part of Figure 7 exemplifies the interlaced pattern between the city and its surroundings, still present long after the forecast period; the region depicted is known as Colônia Santa Luiza, at the south of Ponta Grossa. Old farmlands are surrounded by the city; as a matter of fact, car accidents with farm animals in urban roads, are still not rare (e.g., Globo (2019)).

Some statistics are collected during calibration; the values corresponding to the last iterations are shown in Table 1. The simulator uses a working $\kappa$ that is altered in every iteration. The best value for Matthews' coefficient was found at iteration 25, and $\kappa^*$ was recorded at this point.

A plot of data is shown in Figure 8; curves representing number of pixels were normalized to fit the range $[0; 1]$.

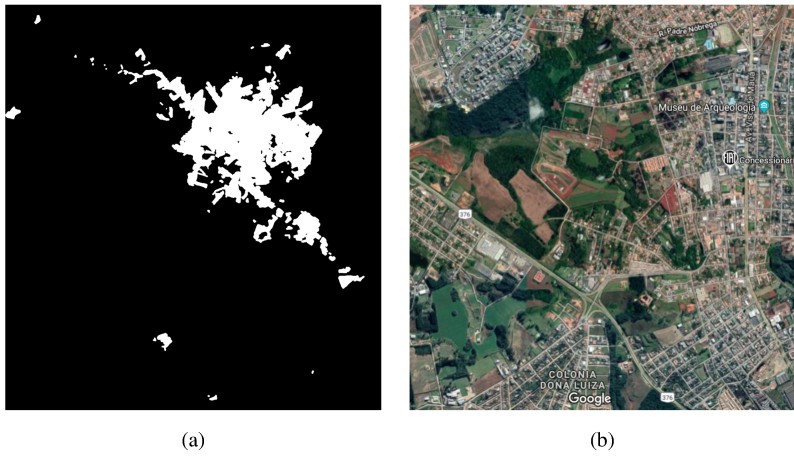

| (a) | (b) |

**Figure 7.** Illustrations of the city morphology; (a) urban footprint 2017 (Roth, 2019); aerial detail 2019 (©Google Earth).

**Table 1.** Calibration for Ponta Grossa.

| Iteration | Heat cycles | True + | True - | False + | False - | Lee-Sallee | Matthews |
|-----------|-------------|--------|--------|---------|---------|------------|----------|
| 23 | 778 | 72098 | 1581179 | 1330 | 8431 | 0.833643 | 0.934622 |
| 24 | 552 | 71459 | 1582093 | 416 | 9070 | 0.857889 | 0.936438 |
| 25 | 636 | 71579 | 1582089 | 420 | 8950 | 0.85661 | **0.937237** |
| 26 | 484 | 71069 | 1582412 | 97 | 9460 | 0.871548 | 0.935959 |
| 27 | 234 | 69863 | 1582509 | 0 | 10666 | 0.879809 | 0.928301 |
| 28 | 234 | 69863 | 1582509 | 0 | 10666 | 0.879809 | 0.928301 |

Approximately 95% of the images are empty space, and this reflects in the fact that the True Negative curve shows little variation.

The results of the UFlow model can be compared with the study developed by Roth (2019), using the SLEUTH simulator and input images that were used in the present work. Three points are noteworthy.

The first aspect is calculation speed; the calibration process in SLEUTH took approximately 40 hours, what is in accordance with remarks of the literature (Clarke-Lauer and Clarke, 2011; Roth, 2019). This delay is dependent on the number of Monte Carlo interactions, but reducing this parameter harms the reliability of results. A series of papers presents a faster version of SLEUTH using Genetic Algoritms, but authors warn that results may present fluctuations (Clarke-Lauer and Clarke, 2011; Clarke, 2017, 2018). By contrast, the calibration of the UFlow model took 10 minutes and is deterministic; random processes
are only active in the prediction step, which runs under a minute.

    The second point is the quantitative results. The SLEUTH simulations in Roth (2019) employed a sequence of images to extrapolate growth in 2017. Results indicated less expansion than expected, and the presumed reason was a phase of slow growth in the sequence of footprints. UFlow follows tendencies found in two input images as a feature of the model, which

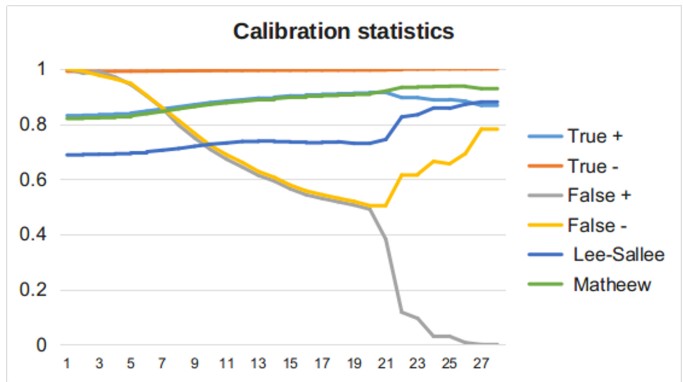

**Figure 8.** Calibration statistics, Ponta Grossa.

translates into more predictable values - although both SLEUTH and UFlow cannot make sure that the growth speed will be
correctly simulated.

The third point is the qualitative aspect of the simulations. SLEUTH tries to assimilate the intensity of different types of expansion across a city. In her study, Roth observed a prevalence of growth around the borders of the images and irregular occurrence of other mechanisms such as expansion along roads. UFlow does not implement a set of growth rules, but, on the other hand, it adapts to the intensity of expansion according to local information. The adjustment to global statistics only occurs
in the generation of scattered fragments.

### 4.1.2  México City

México City occupies a surface of approximately $1485km^2$, with a population of almost 9 million people. The conurbation of the capital of México with neighbour towns exceeds 20 million inhabitants. The city has more than 500 years marked by dramatic events with effects on its morphology.

The city was built on the floor of ancient Texcoco Lake, is surrounded by montains and experiences intense rainfalls. It suffered several floods and in 1629 was covered by water for five years (Martínez, 2004). Systems for drainage were built, but caused ground subsidence. Three tectonic plates meet at the region and are the source of frequent seismic activity. The last notable event was in 2017, with magnitude 7.1 and causing the loss of more than 300 lifes.

Footprints for México City are shown in Figure 9. Data covered the years 1990, 2000 and 2010, with a resolution of 1100
× 1200 pixels. The two decades of city evolution are very distinct. Between 1980 and 1990, urban pixels increased more than 120% (from 14399 to 32527 cells), and the city inflated along most of its perimeter; furthermore, 124 new urban fragments were created. The year 1985 was marked by an earthquake of magnitude 8; it caused approximately 5000 deaths, the collapse of hundreds of buildings and devastated city infrastructures. There was a slight decline of population, although the urban footprint expanded in this period. During the next decade the dynamics were different: between 1990 and 2000, input images showed an



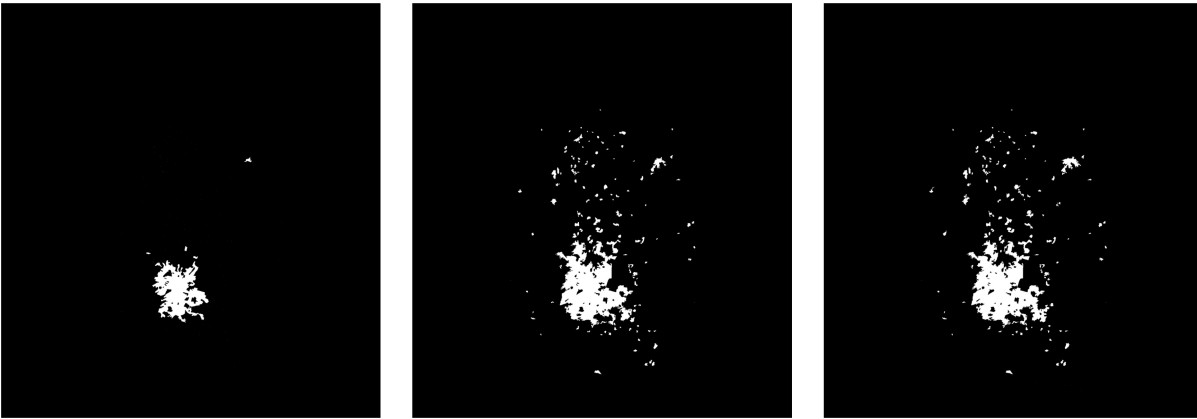

**Figure 9.** México City footprints used in tests; from left to right 1980, 1990 and 2000, images from QGIS plugin SleuthInputs v3.0.

expansion of the urban area of $23\%$ (39989 pixels), and only 2 new fragments appeared. By comparison, population grew only $0.4\%$

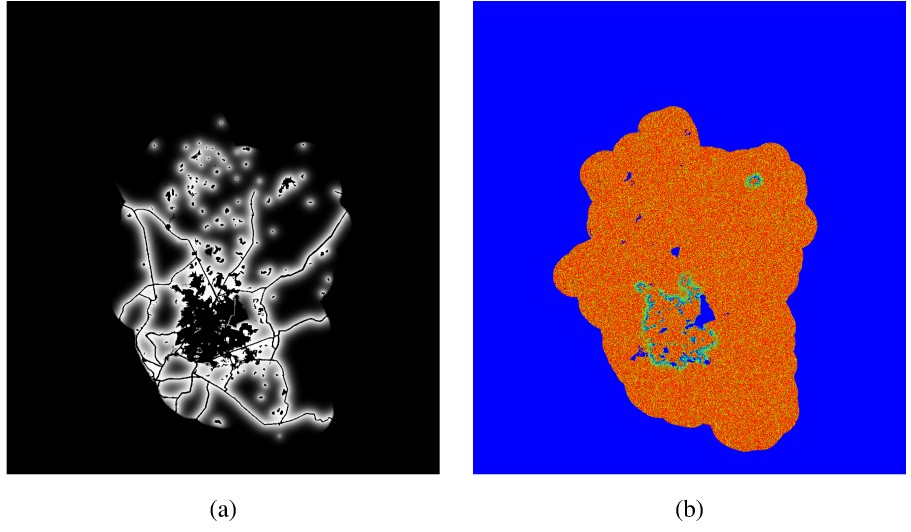

(a)                                    (b)

**Figure 10.** Simulator output: preprocessing and calibration, México City; (a) Distance Map; (b) $\kappa^* + \chi$.

Images computed in the preprocessing and calibration steps are shown in Figure 10. The first part of the figure shows the probability map $\pi$, after removal of areas defined in an exclusion layer, which encompasses satellite towns the compose the Great México conurbation. The $\kappa^*$ map is shown after the addition of uniform noise $\chi \in [-0.25; 0.25]$. This process creates
cracks in the insulation barrier and breaks the regularity of expansion borders. The same method was used in the simulation of Ponta Grossa, but the map in Figure 6 was plotted before noise addition.



Forecasts are shown in Figure 11. The forecast computation is unaware of the events in 1985, which changed the course of the city (López-Cervantes et al., 2014). It predicted the creation of 66 new fragments and augmentation of more than 50% of the total urban area. The expansion speed of a city varies along of the time and may slow down as space becomes scarce

and more expansive. This kind of internal mechanism is not part of the UFlow model, but it can be adjusted by modifying the time spanned by the forecast. It is noteworthy that borders that were sharply defined between 1980/90 remained in place in the forecast and correspond to the actual state of the territory. This is visible in the middle of the city, around a C-shape area near zones classified as risky in the "Atlas de Riesgos de la Ciudad de México".

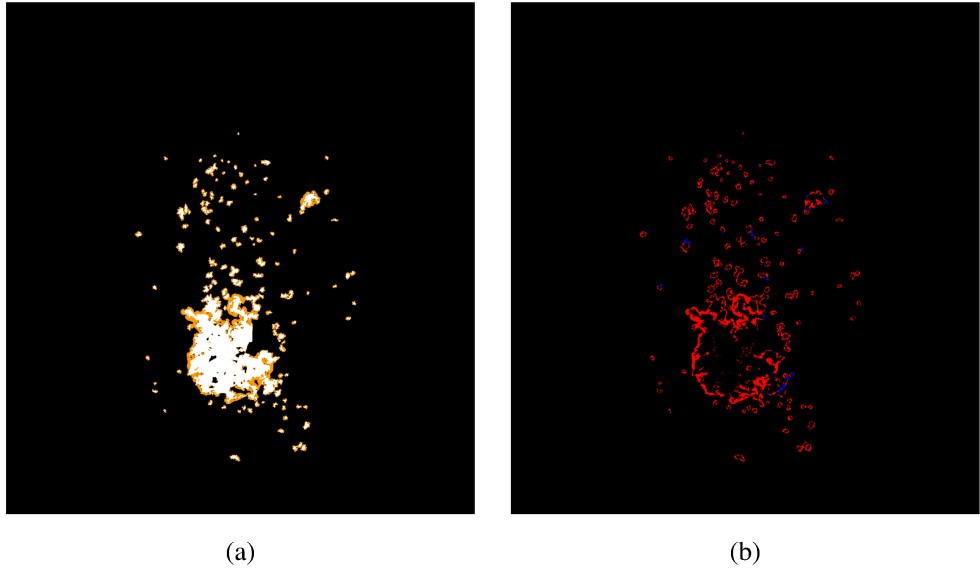

(a)                                  (b)

**Figure 11.** Simulator output: México City; (a) Forecast 2000; (b) Forecast error.

The error image shows a considerable number of false positive pixels, but their distribution is not random. During the

decade 1990-2000, land prices in the south and west areas, where many of those pixels can be found, increased as much as 200% (Parnreiter, 2005), a sign of a strong interest on these zones.

Plots of metrics during the calibration are shown in Figure 12. A salient feature is the sudden drop of False Positive cells, which is the result of the calibration process detecting zones where urbanization did not occur in the simulated period. It might seem paradoxical that the number of False Negatives did not follow the same trend, but this is explained by the fact

that the second image is covered by new fragments, which are not treated during the first phase of simulation. The Lee Sallee and Matthews metrics increase gradually as the calibration proceeds. The metrics present oscillations that are a little more pronounced than in the case of Ponta Grossa, and this effect can be linked to the complex morphology of México City.

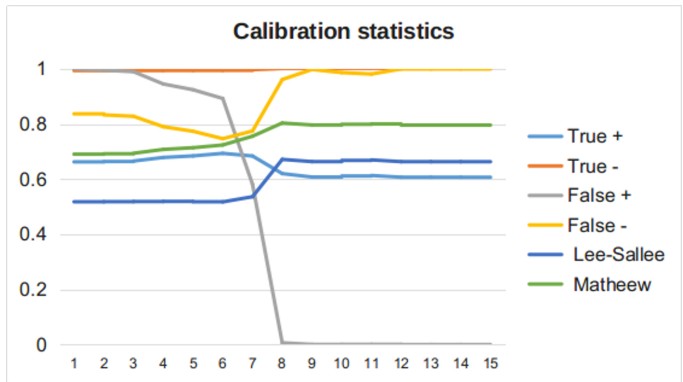

**Figure 12.** Calibration statistics, México City.

## 4.2 Retrograde predictions

A distinctive feature of the UFlow model is the possibility of reversing the direction of time. This is done using negative
images as input and reversing their ordering: the model computes projections about how the empty space enters into the city.
This ability can be explored to compute the approximate state of a location in the past, to have another perspective of analysis
of a region, or to review the calibration of the $\kappa$ map and possibly adjust it.

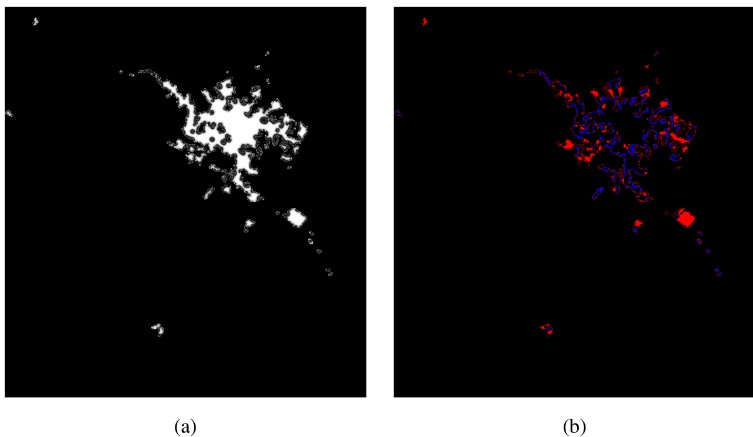

(a)          (b)

**Figure 13.** Simulator output: retrograde computations for Ponta Grossa; (a) Projection 1984; (b) Corresponding error image.

An example for Ponta Grossa is shown in Figure 13.

The random generation of fragments (spraying) was disabled in this experiment; the correct way to perform the reverse sim-
ulation for this case, is to select fragments at random and removing them from the image. This procedure was not implemented,
but that can be done by chosing fragments by colour (painted during cluster detection) and removing them.





After the calculations, the colours of the output image were inverted and the result compared with the 1984 footprint to generate the error image. False positive and negative cells have similar meaning to those used in the forward mode of operation: red pixels should have been removed by the simulation, and the blue ones should have been kept.

The reverse simulation confirmed the tendency found in the forward case (Figure 6), with the same degree of activity accross all the city. The retrograde simulation captured the alternance between static and growing areas along of the whole indented border of Ponta Grossa. The red square that appears at the south of the map corresponds to an industrial zone, which was defined by municipal administration and took form around 1987.

## 5    Conclusions

Simulation of urban growth is not a new theme and the literature provides a body of experience on this subject; the trouble is, despite the methods in existence the problem still stands, and even if the construction of a perfect oracle is un likely goal, the test and development of new approaches help to gain new perspectives and can be a source of elements to build more capable methods.

Among the taxonomies for simulation models, the distinction between data-based and knowledge-based approaches is very common. The proposed model is closer to the former category, putting emphasizing on information as the shape of the urban footprint. At the same time, the conception of the solution is rooted in hypothesis about inner city mechanisms and can accommodate variations as the implementation of rules directly in the CA matrix.

Computing the displacement of city boundaries is not a new idea (Tayyebi et al., 2014); the metaphor of heat flow is able to mimic the phenomenon and also adapt to local conditions. The $\kappa$ matrix distinguishes the model from techniques that calculate
global parameters and apply them indistinctively to the whole area of study.

The section on Verification and Validation shows practical tests with the software, which first confirm the hypothesis of its expected behaviour, and subsequently showed a good agreement with actual data. The model can also be used in reverse mode and make past projections, a feature that was also tested and demonstrated.

There are possibilities for extension. The transportation network is an important element in the generation of scattered
urban fragments, but the relative importance of the roads was not represented in the input data. The assignment of different temperature values is likely to improve the result obtained with the calculation of $\pi = \ell_T(\rho)$. In this case, values in the resulting image must be normalized in the interval $[0; 1]$. Another possible extension to the model is the automatic expansion of the road network (Jiang, 2007; Galin et al., 2011).

The input to the simulator is a binary image that functions as a heat source with uniform values. This information could be
replaced with real estate values, property taxes or a combination of factors composing a normalized urbanization index. The simulation, in this case, might be oriented to compute different results, as prognostics of real estate values, and the calibration procedure redesigned, perhaps with methods from thermodynamics (Alifanov et al., 1995).



The final rendering of the forecast might be improved by applying a post-processing step to the output of $\ell(\cdot)$. The current images exhibit rounded borders that are characteristic of the numerical model, but distinct from the squared shape of city blocks. This might be handled by generating rectangular patches in the regions where expansion occurred.

A distinguishing feature of UFlow is the formulation of an optimization problem. The calibration of the model and the first phase of forecast do not involve stochastic reasoning and the calculations are quite fast. The second phase of forecast - the generation of scattered fragments - is driven by the $\kappa$ map using a sampling (trial-and-error) approach, with a limit of 100 attempts to place each new fragment. According to the logs generated during the tests, this procedure occupied a very small fraction of computing time. Nevertheless, the brute force approach could be compared to the use of an inverse distribution. The pinky function implemented by Tristan Ursell for Matlab might be a good starting point.

Finally, the use of the model to compute retrograde forecasts may help fill gaps in maps depicting the past of a region. Studies aiming the historical reconstruction of archeological sites actually use techniques for simulation of cities, such as Cellular Automata (Ripy et al., 2014), Procedural Generation (Adão et al., 2012) and Virtual Reality (Liu and Tang, 2003).

*Code and data availability.* The implementation is archived with DOI 10.5281/zenodo.3532951. It includes the code, a makefile, user manual and two directories with data. A repository at github is available at https://github.com/AndreKoscianski/UFlow.

*Author contributions.* The author developed and implemented the model. Libraries from other developers are identified in the source code.

*Competing interests.* The author declares that there are no competing interests.



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

## Appendix A: Implementation Details

### A1 Image handling

Data enters the software as image files in the format PNG, Portable Network Graphics. The format is open (there's no copyright), and supports features as transparency, 8 or 16 bits per colour channel, and lossless compression.

The implementation of UFlow includes a copy of the library LodePNG, version 20180611, from Lode Vandevenne. A layer of C++ hides the library and provides utility functions, as adding and subtraccting images, computing intersections and computing metrics as Lee-Sallee and Matthews' coefficient.

A companion application implements these operations separately, as a helping tool for users of the simulator.

### A2 Numerical Integration

Equation (4) is integrated using a first order, forward, finite differences method. Time and space are arbitrarily scaled. Each pixel in the input image corresponds to one node of the mesh, and the distance between nodes is set to 1.0.

The integration method imposes the following stability condition:

$$\kappa \frac{\Delta t}{\Delta x^2} < \frac{1}{2}$$

Since $\kappa \le 1.0$, convergence requires $\Delta t < 2^{-1}$. Smaller values slow down the propagation of the urbanization process, but also makes it more likely that changes in small areas are captured in the updates of $\kappa$.

The core of the integration code is shown in Figure A1.

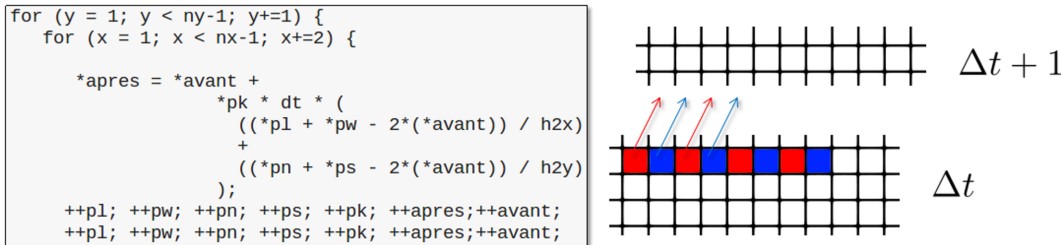

**Figure A1.** Parallelization: code is executed by two threads, pointers take strides between them.

Two threads scan the data array using a stride scheme, represented in the figure by the colours red and blue. Values are
accessed using pointers, making the code a bit faster than using indices. An explanation of the method was found later in (Horak and Gruber, 2005). The scheme can be extended to use more execution threads; modifying the code so that it adapts




**Table A1.** Summary of UFlow parameters.

| Parameter | Access | Typical | Meaning |
|---|---|---|---|
| $T$ in $\ell_T(\cdot)$ | Config file | 4000 | Number of cycles used to compute a Hot Map. |
| $\chi$ | Config file | 0.25 | Magnitude of noise added to $\kappa$ before forecasts. |
| $\gamma$ | Config file | 0.075 | Amount of insulation added at each calibration cycle. |
| $n_C$ | Config file | 30 | Max number of cycles in the outer loop of Algorithm 1. |
| $\Delta t$ | Source code | $2^{-4}$ | Time step in the integration of Equation (4) |
| $\Delta x, \Delta y$ | Source code | 1 | Distance between nodes of the integration mesh. |
| - | Source code | 0.5 | Maximum value for a pixel to be considered non-urban. |
| - | Source code | 5.0 | Temperature of urban footprint, set for calibration and forecast. |

on-the-fly to any number of threads is not difficult either, but that would require pointer arithmetic in place of the increment operator ++, with a slight increase in computational cost. Execution times are of the order of 10 minutes with images of size $1000 \times 1000$ in a laptop with a i3 Intel microprocessor.

The main parameters playing a role in the code are listed in Table A1.

Values were tuned empirically, as new tests unfolded during the implementation. Four parameters were chosen to be made accessible by means of a configuration file. The decision to keep some variables hardwired into the code was rather arbitrary, but had the intention of reducing the complexity the software operation. The exposed variables were deemed sufficient to give control over the simulator.

Researchers interested in modifying the choice of which parameters are accessible will face no special difficulties. The way the configuration file is handled makes it straightforward to add new features. For instance, if a new variable named "Xis" must be added, only two modifications are required:

1. insert a line in the configuration file, as 'Xis = 3.1415';

2. access the variable in the code using this : 'double xis = config_getd ("Xis")'; /* get double */

Other helper functions include config_getb() to read Boolean values, and config_gets() to read strings.

The report system is also simple to extend. A global variable (of type struct SReport) named "GReport" collects all data of interest. The definition of the variable and the generation of the report file are both found in a header file, "report.h". In order to generate the output of a new statistic, for instance a variable named "alpha", the user should:

1. create a new field in the structure, e.g. 'int alpha;';

2. create a new report line: 'arq $<<$ "alpha = " $<<$ GReport.alpha; ';

3. feed the variable in the code this way: 'GReport.alpha = 3.1415';





## Appendix B: Initial tests

figure a and b shows the first tests

they were done x and y

compare with (Hernández-Flores et al., 2017).

compare with Gazulis, N., Clarke, K. C. (2006, September). Exploring the DNA of our regions: Classification of outputs from the SLEUTH model. In International Conference on Cellular Automata (pp. 462-471). Springer, Berlin, Heidelberg.

The first practical tests used images designed to represent different growth patterns, shown in Figure B1; the elapsed time was chosen arbitrarily as 10 years just to provide a reference. The map of roads, slightly non-simmetrical, was already shown

in Figure 4. The simulated city is symmetrical along the horizontal axis. Areas in the southern part remain static, while in the north there are regions growing in patterns of different speed. New isolated fragments appear near edges that did not expand.

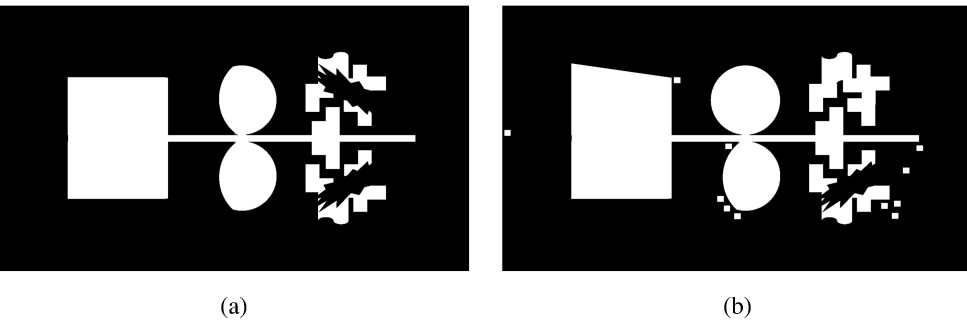

(a)                                              (b)

**Figure B1.** Simulated city maps; (a) year 0 ; (b) year 10.

Outputs computed by the simulator are shown in Figure B2. The first image, B2 a), depicts the $\kappa^*$ coefficient. The red background colour represents maximum conductivity, with $\kappa = 1$ assigned at the beginning of calibration. Blue pixels have value $0$ and function as contention barriers to urban flow. It can be perceived that the whole south part of the city was surrounded

by an insulation barrier.

Image B2 b) shows simulated growth. The simulated city expands in three locations; new pixels occupy a triangle, a D shape, and an irregular pattern. All cases contain different grow gradients, and they were reproduced with good accuracy. Sharp edges in the original image became rounded in the simulation, an expected effect from the behaviour of Equation (4).

The error image B2 c) compares actual data and simulation, showing false positive (red) and false negative (blue) pixels.

Notably, this kind of representation is not used in most studies of urban growth with cellular automata, the literature cited in this paper being an example. These pixels are concentrated where the expansion was faster, or in urban islands that appear only in the second snapshot of the city and are not considered during calibration.

A forecast is shown in Figure B2 d). Before calculating a forecast the simulator modifies $\kappa^*$ with the addition of uniform noise; in this test, this parameter was set to $\chi \in [-0.25; 0.25]$. This gives some margin for growth to occur in areas that were

otherwise inactive in the period spanned between the input images. This procedure also breaks isotherms that form around





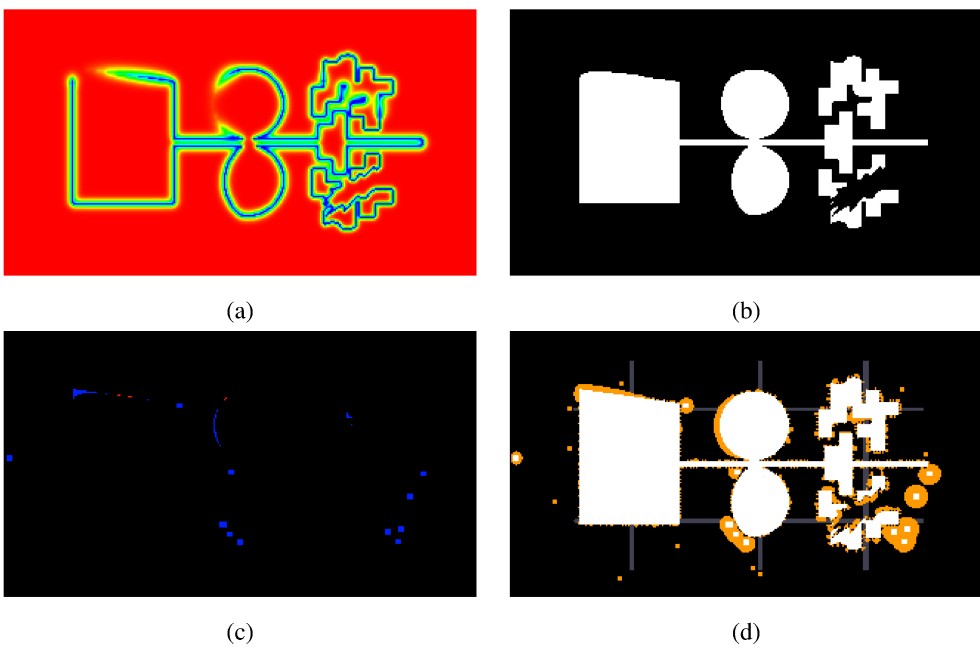

(a)           (b)

(c)           (d)

**Figure B2.** Simulation output for the artificial city; a) $\kappa^*$; b) Simulation year 10; (c) Error image; (d) Forecast year 20

edges and produces a sort of irregular fur, visible in the image. Similar results are visible in the output of other simulators as SLEUTH, and reflect the irregular advance of urban boundaries. Elapsed time was set to the same interval separating the input images. New urban cells are represented as golden pixels, and it can be seen that they appear with more frequency in areas with low insulation. This result shows that every edge of the city may project the urban flow to adjacent areas. Regions where

this should not occur, as parks or lakes, can be defined with the help of an exclusion map.





## Appendix C: Derivation of Heat Equation

Giving a small region $\Delta x$, the flux contained in the region is $\Phi(x + \Delta x, t) - \Phi(x, t)$. Then, the total accumulated over a period of time $\Delta t$ is

$$[\Phi(x + \Delta x, t) - \Phi(x, t)] \Delta t = \kappa \left[ \frac{\partial u}{\partial x}(x + \Delta x, t) - \frac{\partial u}{\partial x}(x, t) \right] \Delta t \tag{C1}$$

By equating expressions (3) and (C1) we obtain

$$[u(x, t + \Delta t) - u(x, t)] \Delta x = \kappa \left[ \frac{\partial u}{\partial x}(x + \Delta x, t) - \frac{\partial u}{\partial x}(x, t) \right] \Delta t$$

, and then, rearranging,

$$\frac{u(x, t + \Delta t) - u(x, t)}{\Delta t} = \kappa \frac{\frac{\partial u}{\partial x}(x + \Delta x, t) - \frac{\partial u}{\partial x}(x, t)}{\Delta x}.$$

Taking the limit when $\Delta t \to 0$ and $\Delta x \to 0$, it comes

$$\frac{du}{dt} = \kappa \left( \frac{\partial^2 u}{\partial x^2} \right)$$

This expression represent diffusion processes in one dimension.