# Peer review of "UFlow 1.0: A Computer Model for Projections of Urban Sprawl"

_Geoscientific Model Development, 2019_

## Referee Comment (RC1) · Martí Bosch (Referee) · 21 May 2020

**1 General comments**

This article presents UFlow, a computational model to simulate urban development as an analogy to the heat diffusion process. The manuscript is overall well executed and I find that it provides a satisfactory overview of how the model works and its potential to represent urban development. Nevertheless, I have two main remarks that I believe would improve the overall quality of the article. Firstly, many technical aspects of the manuscript such as the parameters of the equations or the captions of the figures should be described more clearly. Secondly, the author should further develop the related literature on models of land use/land cover (LULC) change, compare them

with UFlow, and justify in which respect the presented model is an improvement over the others. Finally, since I am not a native English speaker myself, I have not dared to suggest many language corrections, but I believe that the addition of articles is required in several sentences (e.g., "present in heat model" should be "present in the heat model" in line 285)

**2 Specific comments**

**2.1 Technical description of equations, algorithms and figures**

- The author introduces the urbanization index as u(x, y) and then continues as (line 194): "In the one dimensional case and for small values $\Delta x$, $\Delta t$, the amount corresponds to". I do not understand what is meant by "amount". The author should probably replace such word for a more clear description of what the equation 3 below represents. Then, in the line 197, the author writes "Improvements in an area tend to propagate and ultimate lead to urban sprawl". What does "improvements" mean? for the context, I understand that it means an increase of the urbanization index, but I believe that it would be better to write it explicitly. On the other hand, the author should have defined urban sprawl before stating how the increases of the urbanization index relate to it. Similarly, the author follows with "it can be written as" (line 198), but it might not be clear what "it" means in such context. I understand that it refers to the directional flux, but again, it might be better to be more explicit. Likewise it should be stated explicitly that in such analogy, the urbanization index corresponds to the energy. Additionally, the Phi letter used in the equation (line 199) should be described in the following line. Finally, the author states (line 201) "this expression gives rise to the following equation". I understand that this is covered in appendix C, nevertheless, it might be appropriate to make a bit more explicit which transformations lead from one

equation to the other. Additionally, the author should mention that equation 4 is extended to a two-dimensional space.

- The way in which section 3 is structured might be confusing. First of all, I do not understand what the title of sub-section "3.1 Variations on a theme" means. Then, I find strange that the sub-section "3.2 Main algorithm" describes the calibration procedure but omits the generation of new clusters, which I find kind of central to the model. I believe that a better structure would be to have a sub-section describing how forecasts are obtained in UFlow as part of two mechanisms, namely diffusion and generation of new fragments (which I believe corresponds to how the term leapfrog growth is used in the urban sprawl literature). The count of satellite regions could be collapsed into the same sub-section about forecasts, which should be followed by a final sub-section about calibration.

- The figures of section 4 are very incomplete in several respects. In figures 6, 10 and 11, some sort of colorbar would be very helpful, or at least, explaining the ranges of each value in the caption of the figure. I could not find a description of the value ranges of the forecasts (figures 6b and 11a) elsewhere in the manuscript. Similarly, the author should provide a more explicit description of how the error images (figure 6c and 11b) are computed. The terms "Simulator output" used in the captions of figures 10 and 11 are too generic, I believe the author should explain better what each figure shows. On the other hand, some of the color maps employed in such figures might be problematic for readers with common color vision deficiencies. I suggest that the author makes use of a tool such as the ColorBrewer to select better color maps. The captions of figures 8 and 12 should be more informative, and labels for the x and y axes (e.g., iterations and proportion of pixels respectively) must be added. Finally, I believe that the point that the author intends to make with figure 7a should be illustrated differently, because right now it requires the reader to simultaneously look at figures 5, 6c and 7a.

**2.2   Definition of sprawl**

Since UFlow is presented as "a model of urban sprawl" (line 58), the author should define what is understood by the term "urban sprawl" in the context of the UFlow model. Considering how the model works, I suggest escaping the rather futile debate on the exact definition of sprawl (see section 2.1 of doi.org/10.3390/urbansci3020060 - full disclaimer: I am among the authors), but rather define which kind of spatial patterns might be consider as urban sprawl. For instance, in terms of the number of urban patches (or urban islands as used by the author in section 3.3), their size distribution (e.g., numerous small urban patches versus large connected ones) and the like. Such a definition of urban sprawl could be added to the introduction, e.g., in line 29 right after the "several models of urban sprawl".

**2.3   Calibration**

According to the author (and myself), in order to calibrate urbanization models, "a perfect match is of lesser importance if, in a series of snapshots, a model succeeds to reproduce general trends for a city". I find such a view to be inconsistent with the proposed calibration procedure, which is based in a cell-by cell comparison (with the Matthews correlation coefficient as agreement metric). There have been noteworthy developments towards more pattern-based calibration approaches that I believe are more aligned with the foregoing statement (e.g., doi.org/10.1007/s10109-006-0026-9, doi.org/10.1007/s10651-007-0043-y, doi.org/10.1080/13658810210157822, doi.org/10.1016/j.envsoft.2016.04.017 for a review or doi.org/10.1080/13658810410001713399 for epistemological implications when calibrating stochastic LULC models such as UFlow). I understand that incorporating such features would require substantial changes in the software implementation, and thus a first version can be published without them. Nonetheless, I suggest that

the author mentions such an issue either in section 2.2 or in the conclusions, e.g., to discuss the possibility of incorporating a more pattern-based calibration approach in the near future.

**2.4   Comparison with other models**

The sections 2 and 2.1 on LULC change simulation should be extended by reviewing more models and comparing them to the approach proposed in UFlow. Firstly, in the distinction between two types of models introduced in line 75, the author should mention the advantages and shortcomings of each type, as well as explaining to which type of model UFlow belongs and why. Additionally, either in section 2 or in the distinction between cellular automata (CA) models and UFlow of the introduction (line 58), the author should probably mention another key difference between CA and UFlow, which is that in most CA models, rules are explicitly neighborhood-based. Secondly, the author suddenly introduces SLEUTH in section 2.2 (line 27), but I believe that it should instead introduce it explicitly in section 2.1, together with other CA models such as MOLAND (doi.org/10.1016/S0169-2046(02)00218-9) the CLUE model family (https://www.environmentalgeography.nl/site/data-models/data/clue-model/) or FUTURES (doi.org/10.1080/00045608.2012.707591). Given that some of this models are well established in their field, at some point in section 2 (or its sub-sections), section 4 or in the conclusion, the author should emphasize in which respects UFlow is better than the alternatives. This is very briefly done in section 4.1.1 (lines 343-360) but it is restricted to a comparison with SLEUTH.

**2.5   The case for UFlow**

The UFlow model is presented as a "model for projections of urban sprawl", but I believe that the author should justify its potential use in urban planning. The model provides

projections in form of LULC maps which are themselves informative, nevertheless they might lack a practical take-away if planners cannot relate the results to the parameters of the model or tune the parameters to simulate alternative scenarios (see sections 3.3 and 4.1 of doi.org/10.3390/urbansci3020060 for a discussion on this). For example, the residential CA model of Caruso et al. (doi.org/10.1080/13658810410001713371) explores the relationship between the parameters representing residential preferences and the simulated urbanization patterns. In a similar fashion, the patch growing algorithm of the FUTURES model allows the user to encourage infill or sprawl by tuning the value of the "incentives" parameter. In this respect, the author should make the case for UFlow. For instance, how can we use the analogy with heat diffusion to enlighten our understanding of urban sprawl? Or, given its execution speed, should we be using it to run multiple simulations and explore potential bifurcations in the future spatial patterns of urbanization in a way that it would not be possible with the other models? The author briefly comments on this in the conclusion (lines 434-437), but I believe that the practical use of UFlow in urban planning requires further deliberation.

**2.6   Reproducibility**

The source code is openly available at a GitHub repository, however its documentation should be improved in certain respects. First of all, as of commit 902e215, a documentation PDF is found in two different locations of the repository, namely in the root and "source" directories. Regarding the installation documentation, the author does not mention that the Makefile is found in the "source" directory and therefore one should run "make" from there, which actually raises the following error "makefile:5: recipe for target 'all' failed". I finally managed to install it by compiling the code manually, but the author should ensure that the installation steps are properly documented.

**3 Technical corrections**

- Line 18: the author starts with "The expansion of urban areas in many locations of the world is a disorganized process". I believe there are strong grounds to disagree (e.g., "the commuting paradox" doi.org/10.1080/01944369108975516, scaling doi.org/10.1073/pnas.0610172104). It is a complex, largely decentralized process that we hardly understand - but it does not seem "disorganized".

- Line 26: if the reference for Jane Jacobs (note the final "s" in the surname) really corresponds to 1992 (the original book was published in 1961), the edition and publisher should be mentioned.

- Line 29: there is a missing "l" at the end of the term "urban sprawl".

- Line 58: the "using a modifications" should be "using modifications".

- Line 70: the "What's" should be "What is"

- Line 340: figure 8 should be introduced better than as "A plot of data", e.g., "A plot of the evolution of the calibration statistics..."

- Line 369: it is strange to describe the resolution in terms of the number of pixels without providing the the area of the spatial extent (e.g., in hectares).

- Line 403: figure 13 should be introduced better than as "An example for Ponta Grossa" - an example of what, exactly?

---

## Referee Comment (RC2) · Anonymous Referee #2 · 22 May 2020

Report on "UFlow 1.0: A Computer Model for Projections of Urban Sprawl" by Andre Koscianski

The author proposes an urban growth model based on a heat equation spatial-explicitly predicting which raster cells convert to urban. The approach is illustrated by means of an artificial setting as well as the two case-studies Ponta Grossa and Mexico City. The model to some extent correctly predicts historically observed urban growth.

The work seems relevant and might have the potential to be of interest for the specialized community.

However, I cannot recommend publication of the manuscript in the present form. The main reason is incomprehensible presentation. One can hardly understand how the

model works and only guess. Accordingly, the manuscript is lacking a clear description and presentation of the model. What often helps is a step-by-step receipt. I recommend a complete work-over, in particular of the model description.

As the weakness is rather fundamental, it is hard to point out specific problems. Nevertheless, I hope the following details give a hint.

- please provide page of Jacobs-quote (l.24-26) - do not say "very transparent" (l.61), as it is a subjective - all captions are too brief - Eq.(1) seems wrong, ie the set of parameters is equal to the distance metric. I think I understand what the author tries to say, but mathematically it seems wrong. - l.133: one should also distinguish between endogeneous and exogeneous growth - Eq.(3) is not an equation - sort appendices (first to be mentioned is Appendix C) - Fig.3 not clear, what is the temperature? - l.238: what does the superscript "plus" mean? - l.240-250: not clear - l.256: how do new clusters emerge? - l.260: spraying procedure not explained properly? - l.277: what is the distance transform delta()? - l.283: what is g in the exponential? - Algorithm 2: what kind of rectangle? - Fig.6(b) not discussed, skip? - only 1-2 sentences about Fig.8. Either discuss or skip. - computation time (l.345) is not really comparable, because computational power is unknown - maps: scale-bars are missing! - maps: how about surrounding cities/settlements? - why only 2 panels in Fig.10 instead of 3 in Fig.6? consistency! - axis labels of Fig.8 and 12?